# Thermodynamic Driving Forces and Chemical Reaction Fluxes; Reflections on the Steady State

**DOI:** 10.3390/molecules25030699

**Published:** 2020-02-06

**Authors:** Miloslav Pekař

**Affiliations:** Faculty of Chemistry, Brno University of Technology, Purkyňova 118, 61200 Brno, Czech Republic; pekar@fch.vut.cz

**Keywords:** chemical kinetics, chemical thermodynamics, driving force, reaction flux, reaction rate

## Abstract

Molar balances of continuous and batch reacting systems with a simple reaction are analyzed from the point of view of finding relationships between the thermodynamic driving force and the chemical reaction rate. Special attention is focused on the steady state, which has been the core subject of previous similar work. It is argued that such relationships should also contain, besides the thermodynamic driving force, a kinetic factor, and are of a specific form for a specific reacting system. More general analysis is provided by means of the non-equilibrium thermodynamics of linear fluid mixtures. Then, the driving force can be expressed either in the Gibbs energy (affinity) form or on the basis of chemical potentials. The relationships can be generally interpreted in terms of force, resistance and flux.

## 1. Introduction

The thermodynamic analysis of chemically reacting systems is still a lively research area, which continues to include efforts to find deeper relationships between thermodynamics and chemical kinetics. A typical example is the identification of “thermodynamic driving forces”, which could be directly related to reaction rates, usually to rates in both forward and reverse directions, often called fluxes. The driving force is not a precisely defined quantity. The term probably originates with Berthelot [1], who related it to the heat released by a reaction, and for which was soon criticized by Helmholtz. The classical work on the theory of chemical kinetics [2] also includes a brief mention of the driving force, and links it to the strength of the chemical bond formed during a reaction. The energetic nature of the driving force concept has survived to this day [3,4].

Perhaps the most common result of the ‘driving force-reaction flux’ type of relationship is the equation
(1)$$\Delta_{r}G=-RT\ln\left(J^{+}/J^{-}\right)$$
where $\Delta_{r}G$ is the reaction Gibbs energy, or the driving force, and $J^{+}$ and $J^{-}$ are the rates of reaction in the forward and the reverse directions, respectively (the forward and reverse fluxes). Equation (1) can be obtained by directly combining the reaction isotherm equation known from traditional (equilibrium) thermodynamics and the classical reaction rate equation in the form of the mass action law. A crucial point here is the identification of the true thermodynamic equilibrium constant with the kinetic equilibrium constant; this is the main source of certain inconsistencies, which probably cannot be fully resolved even under ideal conditions. More details are given in the review [5] and in the relevant book chapter [6]. In short, the two equilibrium constants are conceptually different. The thermodynamic constant is nondimensional and has no units, while the kinetic constant possesses both dimension and units. The (value of the) thermodynamic constant is dependent on the selection of the standard state, and its dependence on, or independence of a given parameter, e.g., pressure, also depends on this selection (that is, whether the selected standard state depends on the given parameter, e.g., pressure, or not). This is quite strange for kinetic rate constants, which form the kinetic equilibrium constant.

Beard and Qian [7] tried to derive Equation (1) solely on the basis of the conservation of mass, without invoking any rate law, as a fundamental relation for any chemical process operating in the steady state of an open system. The steady state condition thus imposes a certain limitation on their approach. Their argumentation is illustrated in the simple general reaction A⇔B. Let us follow it step by step. First, they suppose that in a nonequilibrium steady state, the numbers of A and B molecules are held constant by A being pumped into the system, and B out of the system. Second, they suppose that A molecules formed by a back-reaction from B molecules can be labeled ($A^{*}$), but are otherwise identical to A molecules. Thus, the following reactions take place formally:(I)A→B
(IIa)$$B\to A^{*}$$
(IIb)$$A^{*}\to B$$

It is further claimed that in a steady state, $A^{*}$ molecules are in equilibrium with B molecules, and that the corresponding equilibrium constant is defined according to $\bar{K}=N_{B}/N_{A^{*}}$, where $N_{i}$ denotes the number of molecules i (in the steady state). This, in fact, means that steps IIa and IIb are taken as a single reversible reaction in equilibrium, which is the essence of another claim that the equality of fluxes is $A^{*}$ → B and B → $A^{*}$. This equality is written in Beard and Qian [7] as
(2)$$J^{+}\left(N_{A^{*}}/N_{\Sigma A}\right)=J^{-}$$
where $N_{\Sigma A}$ is the total number of type A molecules (the sum of A and $A^{*}$). Upon substitution from the definition of $\bar{K}$, we obtain:(3)$$J^{+}/J^{-}=\bar{K}N_{\Sigma A}/N_{B}$$

The thermodynamic definition of the equilibrium constant [7] gives
(4)$$\Delta_{r}{\bar{G}}^{o}=-RT\ln\bar{K}$$
and the corresponding reaction Gibbs energy
(5)$$\Delta_{r}\bar{G}=\Delta_{r}{\bar{G}}^{o}+RT\ln\left(N_{B}/N_{A^{*}}\right)$$
where $N_{i}$ should be taken generally (not just in a steady state).

Combining Equations (3) and (4), we obtain
(6)$$\Delta_{r}{\bar{G}}^{o}+RT\ln\left(N_{B}/N_{\Sigma A}\right)=-RT\ln\left(J^{+}/J^{-}\right)$$
which, however, is not Equation (1) for the exemplified reaction. For this, Equation (1) should read
(7)$$\Delta_{r}G^{o}+RT\ln\left(N_{B}/N_{\Sigma A}\right)=-RT\ln\left(J^{+}/J^{-}\right)$$
where $\Delta_{r}G^{o}=-RT\ln K$ and $K={\left(N_{B}/N_{\Sigma A}\right)}_{\mathrm{eq}}$. This discrepancy, mixing K and $\bar{K}$, seems to have gone unnoticed.

In this work, some different—let us say traditional—approaches to steady state kinetics are analyzed from the ‘driving force-reaction flux’ point of view, and supplemented with some more general results of the nonequilibrium thermodynamics of chemically reacting mixtures.

## 2. Results and Discussion

### 2.1. Steady States of Basic Open Reacting Systems

In chemical kinetics (chemical reaction engineering), a steady state is a clear concept referring to a special state of open systems (reactors). Two principal models of open systems are used: the well-mixed system with continuous input and output (the continuous stirred tank reactor or CSTR); and the tubular flow-through system with a plug flow regime (the plug flow reactor or PFR). Let us analyze the steady states of these systems for the above simple reaction from the point of view of driving force-flux relationships. This basic analysis is still missing in the area of seeking these relationships for chemically reacting systems. First, it should be noted that PFR can be modeled as a series (of a sufficient number) of CSTRs; thus, here we will only focus on the CSTR system. As in [7], we suppose the validity of the following equation, the reaction isotherm originating in classical (equilibrium) thermodynamics:(8)$$\Delta_{r}G=-RT\ln K+RT\ln Q\equiv \Delta_{r}G^{o}+RT\ln Q$$

Here, $K=c_{Be}/c_{Ae}$ is the equilibrium constant (for an ideal system with a standard state of unit concentration; subscript *e* denotes equilibrium) and $Q=c_{B}/c_{A}$ is the reaction quotient, where $c_{i}$ is the concentration of i. Obviously, the component reaction rates fulfill the condition: $J_{A}=-J_{B}=-J$ (all in mol m^−3^ s^−1^).

The molar balance of CSTR (Figure 1) with inputs $F_{i}^{0}$ (referred to as “pumping”, especially in bio-related works) and outputs $F_{i}$ (both in mol m^−3^ s^−1^) in a steady state (no special index is used to denote steady state values) is given by
(9)$$F_{i}^{0}+J_{i}=F_{i}=c_{i}\left(V_{F}/V\right)=c_{i}/\tau$$
where $V_{F}$ is the volumetric flow rate (m^3^ s^−1^) and V, the reactor (system) volume, τ is usually called the space time. From the balance (Equation (9)), the steady state concentrations can be expressed as follows:(10)$$c_{A}=\tau \left(F_{A}^{0}-J\right),\;c_{B}=\tau \left(F_{B}^{0}+J\right)$$

The reaction quotient in the steady state is then:(11)$$Q=\left(F_{B}^{0}+J\right)/\left(F_{A}^{0}-J\right)$$

Equilibrium can be defined as a state in which J=0. Then, from (10), $c_{Ae}=\tau F_{A}^{0}$ and $c_{Be}=\tau F_{B}^{0},$ and consequently, $K=F_{B}^{0}/F_{A}^{0}$. However, the last equation cannot be seen as a definition of the equilibrium constant, but rather as a condition on the setting of the reactor input to attain equilibrium. In fact, the input composition is then identical to the equilibrium composition, and no reaction occurs in the system (it does not even begin).

Introducing Equation (10) into Equation (8), we obtain
(12)$$\Delta_{r}G-\Delta_{r}G^{o}=RT\ln\left[\left(F_{B}^{0}+J\right)/\left(F_{A}^{0}-J\right)\right]$$
and
(13)$$D\equiv \exp\left[\left(\Delta_{r}G-\Delta_{r}G^{o}\right)/RT\right]=\left(F_{B}^{0}+J\right)/\left(F_{A}^{0}-J\right)$$
where D is a measure of the distance (of an actual state described by the value of $\Delta_{r}G$) from equilibrium. That is, the distance from the standard state. However, the standard state is intimately related to the equilibrium descriptor through the definition $\Delta_{r}G^{o}=-RT\ln K$. In fact, D is equal to the reaction quotient; however, we will use a special symbol to highlight its definition by the identity in Equation (13), while the reaction quotient is regularly written as a proper fraction with concentrations (activities). For example, see $Q=c_{B}/c_{A}$ above. The reaction rate can be expressed from Equation (13) explicitly
(14)$$J=\left(DF_{r}^{0}-1\right)F_{B}^{0}/\left(D+1\right)$$
where $F_{r}^{0}=F_{A}^{0}/F_{B}^{0}$, and denotes the ratio of the components in the input.

Equations (13) and (14) are the most general relationships which can be derived from the basic equations (balances) describing this system. On the left-hand side of Equation (13), we can see the thermodynamic forcing in terms of the distance from equilibrium. On the right-hand side, we can see the kinetic forcing (the composition of the input) together with the kinetic outcome: the reaction rate. This result should be understood much more like the “bookkeeping” of the (steady state) situation in the system than as some predictor-like equation. Of course, the reaction rate can be expressed from this bookkeeping, as seen in Equation (14). Here again, the rate is related to the thermodynamic forcing (D), together with the (macroscopic) kinetic forcing, which is simply the input composition ($F_{i}^{0}$). Selected examples for $F_{B}^{0}=1$ are shown in Figure 2. It should be remembered that even the thermodynamic forcing is determined by the composition, as $\Delta_{r}G$ can be expressed in terms of chemical potentials, which in turn can be expressed in terms of concentration.

In equilibrium, $F_{r}^{0}=1/K$, $D_{e}=K$ and Equation (14) predict a zero reaction rate, as expected. When there is no B in the input, Equation (13) gives the following simplified version of Equation (14)
(15)$$J=DF_{A}^{0}/\left(D+1\right)$$
which predicts a positive reaction rate, again as expected.

It is interesting to compare a flow-through system with a batch (closed) system (reactor). Here, there is no steady state; the reaction reaches equilibrium, where the reaction rate is zero. The general (nonstationary, i.e., out of equilibrium) balance at constant volume is $dc_{i}/dt=J_{i}$. In contrast to the previous CSTR steady state example, the rate is not constant or related directly to concentration. Integrating the balance, we obtain:(16)$$c_{i}-c_{i}^{0}=\int_{0}^{t}J_{i}dt\equiv I_{i}$$

Because $J_{A}=-J_{B}$, $I_{A}=-I_{B}\equiv -\int_{0}^{t}Jdt\equiv -I$, and:(17)$$c_{A}=c_{A}^{0}-I,\;c_{B}=c_{B}^{0}+I$$

Thus,
(18)$$Q=(c_{B}^{0}+I)/\left(c_{A}^{0}-I\right)$$
and
(19)$$\Delta_{r}G=-RT\ln K+\;RT\ln\left[\left(c_{B}^{0}+I\right)/\left(c_{A}^{0}-I\right)\right]$$

The distance measure is here again equal to Q:(20)$$D=\left(c_{B}^{0}+I\right)/\left(c_{A}^{0}-I\right)$$

Then we have the final “force-flux” relationship
(21)$$I=\left(Dc_{A}^{0}-c_{B}^{0}\right)/\left(D+1\right)=\left(Dc_{r}^{0}-1\right)c_{B}^{0}/\left(D+1\right)$$
($c_{r}^{0}=c_{A}^{0}/c_{B}^{0}$). As noted above, here we do not see the actual (instantaneous) reaction rate, but its integral (I) up to a specific time. Nonetheless, in Equation (21), we see again the thermodynamic forcing, D, related to this integral kinetic characteristic, together with the kinetic forcing, which in this case is determined by the initial concentrations of both components. If there is no product B present at the beginning:(22)$$I=Dc_{A}^{0}/\left(D+1\right)$$

In equilibrium—which is the final state here, in contrast to the steady state—we have:(23)$$I=\left(Kc_{A}^{0}-c_{B}^{0}\right)/\left(K+1\right)$$

These results, arising from the standard balances of a specific reacting system (reactor), can be summarized in the following way. Though a reaction has a unique reaction Gibbs energy, its relationship to kinetics depends on the system in which the reaction occurs. This is due to the concentration dependence of the reaction Gibbs energy and the different equations balancing the concentrations in different systems. Nevertheless, the two “flux-force” relationships of Equations (14) and (21), or Equations (15) and (22), are analogous, as the rate characteristics (J or I) are related to the thermodynamic “distance” (D) and to the initial state, be it the starting time or the input into the reaction system. Both relationships include the history of the reaction up to a specific time point. In the case of the flow-through system, this is (the steady-state point and is) represented just by the steady-state reaction rate, i.e., a single value, which is, however, the result of the previous development of the reacting system before a steady state had been achieved. In the case of a batch system, there is no single specific rate value, and the history is expressed by the integral from the initial to the actual time. Note that the “flux-force” relationships of Equations (14) and (21), or Equations (15) and (22), do not require the separation of the reaction rate into forward and reversed rates, nor do the reaction rates need to be expressed in a mass-action form.

### 2.2. Steady State as a Kinetic Approximation

The steady state has yet another connotation in chemical kinetics. It is used also as an approximating tool in the description of complex reaction schemes, stating that the concentration of (reactive) intermediates is constant (and usually very low). The analyzed single reaction is too simple for the application of this tool, and the easiest extension is the following two-step scheme
(R1)A+C=AC
(R2)AC=B+C
in which AC is the intermediate and C can be viewed, for example, as a catalyst accelerating A to B conversion. Let us start with the batch system, which will provide results quickly and easily. The balances are
(24)$$\begin{array}{c} dc_{A}/dt=-J^{1}\\ dc_{B}/dt=J^{2}\\ dc_{C}/dt=-J^{1}+J^{2}\\ dc_{\mathrm{AC}}/dt=J^{1}-J^{2}=-dc_{C}/dt \end{array}$$
where $J^{1}$ is the rate of (R1) and $J^{2}$ the rate of (R2). The steady state approximation for AC gives $dc_{\mathrm{AC}}/dt=0,$ from which it also follows that $dc_{C}/dt=0$ and $J^{1}=J^{2}$. Thus, $J_{A}=-J_{B}$ and the same situation results in the simple transformation A=B above. Note that without the steady state approximation, $J_{A}\neq -J_{B}$ generally, but $J_{A}=-J^{1},\;J_{B}=J^{2}$.

The nonstationary CSTR balance for AC in the scheme (R1)–(R2) is:(25)$$F_{\mathrm{AC}}^{0}+J^{1}-J^{2}=dc_{\mathrm{AC}}/dt+F_{\mathrm{AC}}$$

Because AC is an intermediate formed inherently during the reaction, it is quite reasonable (perhaps even necessary) to assume that it has no input or output. The steady state approximation then gives $J^{1}-J^{2}=0$. Let us denote the common value of the two reaction rates as J; the remaining balances are then:(26)$$\begin{array}{c} F_{A}^{0}-J=dc_{A}/dt+F_{A}\\ F_{B}^{0}+J=dc_{B}/dt+F_{B}\\ F_{C}^{0}=dc_{C}/dt+F_{C} \end{array}$$

Thus, the accumulation of C is only due to the difference between its input and output. In the reactor steady state, the time derivatives in Equation (26) vanish and the analysis given above for the simpler case of A=B remains valid, as the last Equation (26) does not affect the first two Equations (26). Note that if the “catalyst” C is also not pumped into or out of the reactor, the steady state approximation is automatically valid for this as well.

### 2.3. An Alternative Offered by Non-Equilibrium Thermodynamics

Yet another quite general approach to the “flux-force” topic is provided by continuum nonequilibrium thermodynamics, as developed by Samohýl for chemically reacting mixtures of linear fluids [8]. Linear fluids appear in many systems commonly encountered in chemistry. In comparison to the basic analysis in the previous parts, nonequilibrium thermodynamics provides quite general conclusions that are restricted neither to specific reactor systems nor to steady states. The theory is explained in the referenced book, and here, only results relevant for the discussed topic and example are given. First, the theory derived what had for a long time been a matter of empirical knowledge in phenomenological kinetics [9]. Namely, the fact that the reaction rate is generally a function of (only) temperature and concentration:(27)J=J(T, c)

Here, J is the vector of reaction rates whose components are the rates of R independent reactions, $J=\left(J_{1},\;J_{2},\ldots ,\;J_{R}\right),$ and c is the vector of molar concentrations. The function is not dependent on the reactor type in which reactions occur, nor is it limited to a steady state. Note that the theory generally does not need the assumption that the reaction rate is the difference between the forward and backward rates. To find some “flux-force” relationship, it is necessary to transform this function to a function of Gibbs energy, or equivalently, of affinity. Samohýl’s method takes two important facts into account: first, the transformation should be mathematically correct; and second, the transformation to affinity should conform to stoichiometric constraints.

Function (27) can be easily transformed into a function of chemical potentials, if we accept the widely used relationship between concentration and chemical potential (in ideal systems): $\mu_{i}=\mu_{i}^{o}+RT\;\ln c_{i}$ (unit standard concentration used but not emphasized). It should be understood that this is a rather specific formula, which simplifies the general finding that the chemical potential of a component is a function of the concentrations of all components. The standard chemical potential $\mu_{i}^{o}$ is not dependent on concentration. We thus have:(28)J=J(T, c)→J=J(T, μ)

As shown by Bowen [10], the linear algebra of reaction stoichiometry results in the conclusion that the vector of chemical potential generally decomposes into two perpendicular vectors
(29)μ=A+B
where vector A is the vector of chemical affinities or reaction Gibbs energies (its components are $A^{p}=\sum_{i=1}^{n}\mu_{i}P^{pi}=\Delta_{r}G^{p},$ where p refers to the independent reaction, $P^{pi}$ is the stoichiometric coefficient of the component i in reaction p and n is the number of components (In the book by Pekař and Samohýl [8] the affinity is defined by the reversed sign following the tradition of classical thermodynamics.) and vector B is the vector of constitutive affinities [5]. Using the decomposition (Equation (29)), the final transformation is given by:(30)J=J(T, μ)→J=J(T, A, B)
Thus, the reaction rate cannot be expressed as a function of chemical affinity only. In other words, it is a function of a single affinity (or Gibbs energy). The constitutive affinity reflects the atomic composition of the components of the reacting mixture through its specific combination of chemical potentials [5,8]. The second equation in Equation (30) can thus be interpreted as pointing to the fact that not only the chemical potentials of the reactants and products (which are combined in A) but also the way in which atoms combined in the reactants and products (which is reflected in B) participate in the driving force.

Let us demonstrate the transformation just on the discussed example of a single reaction from introduction A⇔B. The full derivation of all equations is given in our recent work [11], and only the results important for this tractate are presented here. There is only one independent reaction with the (first order) rate equation [11]:(31)$$J=k_{1}\left(c_{A}-K^{-1}c_{B}\right)$$

Its transformation to the function of affinities is given by:(32)$$J=k_{1}\exp\left(-\mu_{A}^{o}/RT\right)\exp\left(B/RT\right)\exp\left(A/2RT\right)\left[\exp\left(-A/RT\right)-1\right]$$

The italicized symbols A and B represent the two affinities (not the two components). Equation (32) shows a rather complex relationship between the reaction rate and the (thermodynamic) driving force in which both affinities are involved. It can be modified to the following “condensed” form:(33)$$J=k^{o}\exp\left[\left(A+2B\right)/2RT\right]\left[\exp\left(-A/RT\right)-1\right]$$
($k^{o}=k_{1}\exp\left(-\mu_{A}^{o}/RT\right)$). Equations (32) and (33) are analogs of Equation (1), which could be rewritten in the form $J^{+}/J^{-}=\exp\left(-\Delta_{r}G/RT\right)\equiv \exp\left(-A/RT\right)$, and show that both affinities are involved in what could be called the “driving force” in Equations (32) and (33). Besides affinities (thermodynamic factors), these equations also contain the kinetic factor ($k_{1}$), as should be expected. Perhaps the expression in square brackets (the second in the case of Equation (33)) could be called the “principal” or “leading” driving force, as it is this expression which ensures a zero reaction rate in equilibrium (where A=0) and resembles the expression $\exp\left(-\Delta_{r}G/RT\right)$, appearing in Equation (1), which was the motivation behind this paper.

The two affinities in this example are as follows [11]: (34)$$A=-\mu_{A}+\mu_{B};\;B=\left(1/2\right)\mu_{A}+\left(1/2\right)\mu_{B}$$

Thus, $A+2B=2\mu_{B}$ and the first exponential in Equation (33) contain only the chemical potential of B (the product). This is natural, as this very simple reaction represents some isomerization, and the same atoms are still combined in the isomers. Consequently, the role of the constitutive affinity in the driving force mentioned after Equation (30) above is not so important here.

This approach does not require forward and reversed reaction rates either.

### 2.4. Chemical Potential in the Role of the Driving Force

An alternative search for the reaction driving force can focus on chemical potentials and not just on the Gibbs energy. The method involving the continuum nonequilibrium thermodynamics of linear fluid mixtures gives the following equation for the discussed reaction:
(35)$$\begin{array}{l} J=k_{1}\left[\exp\left(\frac{\mu_{A}-\mu_{A}^{o}}{RT}\right)-\exp\left(\frac{\mu_{B}^{o}-\mu_{A}^{o}}{RT}\right)\exp\left(\frac{\mu_{B}-\mu_{B}^{o}}{RT}\right)\right]\\ \;\;\;\;\;\;\;\;=k_{1}\left[\exp\left(\frac{\mu_{A}-\mu_{A}^{o}}{RT}\right)-\exp\left(\frac{\mu_{B}-\mu_{A}^{o}}{RT}\right)\right]\\ \;\;\;\;\;\;\;\;=k_{1}\exp\left(\frac{-\mu_{A}^{o}}{RT}\right)\left[\exp\left(\frac{\mu_{A}}{RT}\right)-\exp\left(\frac{\mu_{B}}{RT}\right)\right]={\bar{k}}^{o}\left[\exp\left(\frac{\mu_{A}}{RT}\right)-\exp\left(\frac{\mu_{B}}{RT}\right)\right] \end{array}$$

The term in the last square brackets can be viewed as the (thermodynamic) driving force, which is simply and naturally rooted in the difference in the (exponentials of the) chemical potentials of the reactants and products. If the chemical potential of reactant (A) is higher than that of the product, the reaction rate is positive, as required. When the two potentials are equal, the reaction rate vanishes, which is the equilibrium state. Note that Equation (35) still contains the kinetic factor (${\bar{k}}^{o}$). Very similar conclusions were reported for a simple enzymatic reaction [12]. Efforts to formulate the thermodynamic driving force in terms of energy are probably motivated by the general use of energy in the descriptions of the causes and directions of natural processes. Natural processes are usually described as being “propelled” by a high energy, running in the direction of decreasing energy and settling at the point of minimum energy. Perhaps chemical reactions are better described in terms of chemical potential, which is a (compositional) derivative of energy, and chemical reactions are processes changing the composition.

## 3. Conclusions

Steady states of two basic open systems of traditional chemical kinetics were analyzed from the point of view of the thermodynamic force-reaction flux perspective. Further, a general view on this force-flux issue was presented, which follows from continuum nonequilibrium thermodynamics of chemically reacting linear fluids.

It seems important to differentiate between the identification of a thermodynamic driving force for a chemical reaction and the finding of a relationship between this driving force and the reaction rate. Such relationships should also contain—besides the driving force and general parameters like temperature or the universal gas constant—some kinetic factor(s).

The driving force behind a chemical reaction can probably be seen in terms of the difference between the energetic states of its reactants and products. Relating the driving force to the reaction rate (“flux”) was achieved by combining the concentration dependencies of both the force and the rate. Thus, there need not be a single universal “flux-force equation”, but rather diverse relationships for specific concentration dependencies and reacting systems. In other words, these are not expressions of the cause-effect type, but rather the results of the transformations of functions (i.e., the results of changes of independent variables), and express a form of “bookkeeping” in a reacting system.

The relationships derived from the nonequilibrium thermodynamics of linear fluids, Equations (32) or (35), could be generally interpreted in the following form:(36)$$\mathrm{reaction}\;\mathrm{rate}=\frac{\mathrm{driving}\;\mathrm{force}}{\mathrm{resistance}}\;\;\mathrm{or}\;\mathrm{flux}=\frac{\mathrm{force}}{\mathrm{resistance}}$$
The resistance, or more properly, its reciprocal value, represents the kinetic factor, while the driving force represents the thermodynamic (energetic) factor. The thermodynamic factor can be formulated in terms of the reaction Gibbs energy (equivalently, affinities) or the chemical potentials of the reactants and products.

## Figures and Tables

**Figure 1 molecules-25-00699-f001:**
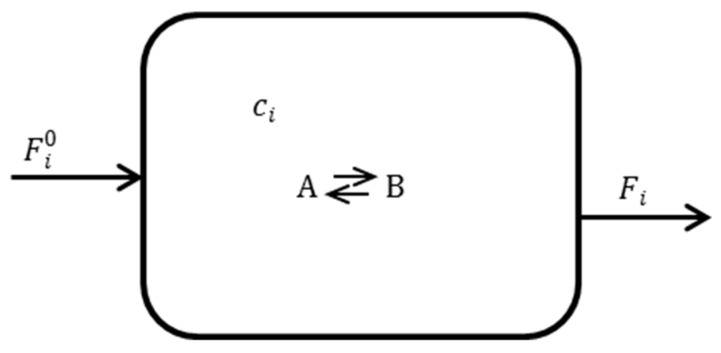
Scheme of the continuous stirred tank reactor (CSTR).

**Figure 2 molecules-25-00699-f002:**
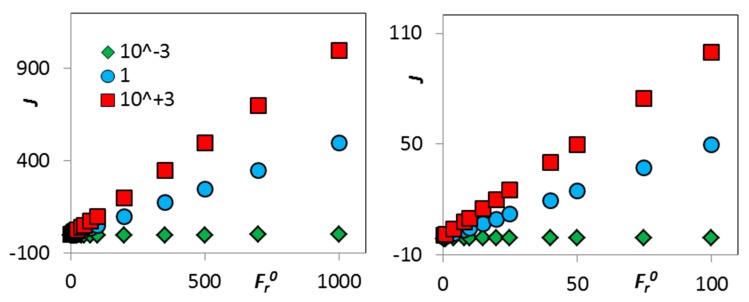
Schematic examples of the dependence of the reaction rate on the kinetic forcing ($F_{r}^{0}$) for various values of the thermodynamic forcing (D) shown in the legend; Equation (14), $F_{B}^{0}=1$. Left: overall view, right: detailed view.

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
