# Peer review of "Thermodynamic Driving Forces and Chemical Reaction Fluxes; Reflections on the Steady State"

_molecules, 2020, doi:10.3390/molecules25030699_

Round 1
Reviewer 1 Report
I have been asked to review the following manuscript, entitled:
"Thermodynamic driving forces and chemical reaction fluxes; reflections on the steady state"
submitted for publication as a regular article in Molecules.
The manuscript explores relationships between the so-called thermodynamic driving force and the reaction rate for simple chemical reactions. Special attention is given to steady state conditions. It is argued that such relations should contain, besides standard parameters like for example temperatures, etc, also kinetic factors characteristic of the reactions. The manuscript is indeed well written and pedagogically presented. The results are interesting and the calculations are convincing. The manuscript indeed discusses an interesting fundamental problem on chemical kinetics. I only have few points that must be addressed:
1) In line 110 (Page 3), one concludes that K=1. So, this must be a typo.
2) In Fig. 2, it looks like you are referring to Eq. (14) and not to Eq. (8). Could you please clarify this point?
3) In line 247 (Page 7), it appears letter alpha in P's superscript. This letter is out of place and should be replaced by letter i.
In conclusion, I recommend the publication of this manuscript in Molecules.
Author Response
1) In line 110 (Page 3), one concludes that K=1. So, this must be a typo.
Yes, this was missprint. Corrected. Thanks.
2) In Fig. 2, it looks like you are referring to Eq. (14) and not to Eq. (8). Could you please clarify this point?
Yes, this was another missprint. Corrected. Thanks.
3) In line 247 (Page 7), it appears letter alpha in P's superscript. This letter is out of place and should be replaced by letter i.
You are right. Corrected. Thanks.
Reviewer 2 Report
The author analyzed the nonequilibrium properties of simple chemical reactions at steady state regimes through two distinct approaches. Both of them take into account kinetic equations combined with gibbs energy relations in order to connect to the thermodynamics. Although the metodology seems to be correct, the way the manuscript was written does no allow me to judge its novelty. In addition, two distinct approaches ( steady states of basic reacting systems and continuous approach) were analyzed and no comparison between them was performed. What is more reliable? What are their advantages/disadvantages? All these questions make me unable to recommed the manuscript for publication. I am willing to examine the manuscript again, provided the author performs an extensive revision of manuscript and stating above points under a much clearer way. Other points that should me clarified/improved. 1)in 110th line, it was mentioned that "the last equation cannot be seen as a definition of the equilibrium constant, but rather, as a condition on the setting of the input to attain equilibrium". Do not these issues strongly related? 2)Do the data in figure 2 come from Eq. (14) or numerical simulations of chemical reactions?A comparison should be performed in order to validate obtained relations. 3)In 234 th line it was mentioned that "the transformation should me mathematically correct,.."since this is what expectes from every tranformations, such sentence does mot make sense. 4) how about the author evaluate the entropy production and allied quantities, in order to strenghten results and the nonequilubrium models features? 5)Conclusions have to be rewritten in order to state clearly the main goals and findings of the paper. Summarizing, the manuscript is not recommended for publication.
Author Response
The author analyzed the nonequilibrium properties of simple chemical reactions at steady state regimes through two distinct approaches. Both of them take into account kinetic equations combined with gibbs energy relations in order to connect to the thermodynamics. Although the metodology seems to be correct, the way the manuscript was written does no allow me to judge its novelty. In addition, two distinct approaches ( steady states of basic reacting systems and continuous approach) were analyzed and no comparison between them was performed. What is more reliable? What are their advantages/disadvantages?
In Introduction some discrepancies in a previous analysis of driving forces in steady state were mentioned. Because steady state has a clear meaning in chemical kinetics and driving forces have not been analyzed under this meaning, the first parts of the manuscript are devoted to this issue as stated at the end of Introduction. The continuum non-equilibrium thermodynamic approach was further added because it is more general and not restricted to steady states. Text was modify to explain these aims.
1)in 110th line, it was mentioned that "the last equation cannot be seen as a definition of the equilibrium constant, but rather, as a condition on the setting of the input to attain equilibrium". Do not these issues strongly related?
Yes, they are related, nevertheless the reported (last) equation is not a definition.
2)Do the data in figure 2 come from Eq. (14) or numerical simulations of chemical reactions? A comparison should be performed in order to validate obtained relations.
Figure 2 simply illustrates Eq. (14), the misprint in the figure legend was corrected.
3)In 234 th line it was mentioned that "the transformation should me mathematically correct,.."since this is what expectes from every tranformations, such sentence does mot make sense.
You are right but there are cases in literature which do not care about mathematics of inversions of functions or changing variables and change them at will. This sentence should stress the correctness which is illustrated in the subsequent paragraph.
4) how about the author evaluate the entropy production and allied quantities, in order to strenghten results and the nonequilubrium models features?
Actually, entropy inequality was one of basic principles upon which was built the theory used in parts 2.3 and 2.4. The theory was explained in ref. 8 as noted in the beginning of part 2.3 (due to its length it cannot be reproduced in papers using it). Thus, principal consequences of entropy inequality are already contained in the theory as is used in parts 2.3 and 2.4.
5)Conclusions have to be rewritten in order to state clearly the main goals and findings of the paper.
The main goals were restated. In my opinion, conclusions states just the main findings.
Round 2
Reviewer 2 Report
The manuscript has been partially revised by the author. Although there is some improvement and the subject is interesting, in my opinion it does not contain enough results to deserve publication. In particular, results of figure 2 should be compared with numerical simulations. Also, an analysis in terms of entropy production should be performed. These issues have been pointed out previously and the author did not answered in the first report.
Author Response
In particular, results of figure 2 should be compared with numerical simulations.
The figure just shows numerical simulations - it just graphically illustrates outputs of equation (14) for a reader to have an immediate pictorial idea of what the outputs can be, of what flux-force relationships can look like in the discussed example. The equation, as shown in the manuscript, was derived from the steady-state balance equations of the continuous stirred tank reactor, thus, the equation simulates the discussed reaction generally.
Also, an analysis in terms of entropy production should be performed. These issues have been pointed out previously and the author did not answered in the first report.
In my first report it was stated: "Actually, entropy inequality was one of basic principles upon which was built the theory used in parts 2.3 and 2.4. The theory was explained in ref. 8 as noted in the beginning of part 2.3 (due to its length it cannot be reproduced in papers using it). Thus, principal consequences of entropy inequality are already contained in the theory as is used in parts 2.3 and 2.4." The entropy production is irrelevant for the discussion of the steady state behavior because all required results are obtained from the steady state reactor balances as shown in the manuscript. In other words, these results can be viewed as an addition to outcomes of the entropy production.